# Use of Knotless Barbed Sutures in Laparoscopic Inguinal Hernioplasty in Horses: 40 Cases

**DOI:** 10.3390/ani14121826

**Published:** 2024-06-19

**Authors:** Francisco J. Vázquez, David Argüelles, Juan A. Muñoz, Martin Genton, José L. Méndez Angulo, Frederic Climent, Imma Roquet, Manuel Iglesias, Ana Velloso Álvarez, Arantza Vitoria, Fernando Bulnes, Aritz Saitua, Antonio Romero, Javier Ezquerra, Marta Prades, F. Javier López-Sanromán, Fabrice Rossignol

**Affiliations:** 1Servicio de Cirugía y Medicina Equina del Hospital Veterinario, Departamento de Patología Animal, Facultad de Veterinaria, Universidad de Zaragoza, 50013 Zaragoza, Spain; pvazquez@unizar.es (F.J.V.); avm@unizar.es (A.V.); aromerol@unizar.es (A.R.); 2Hospital Clínico Veterinario, Universidad de Córdoba, 14014 Córdoba, Spain; fbulnesjm@gmail.com (F.B.); aritz_sp91@hotmail.com (A.S.); 3Departamento de Medicina y Cirugía Animal, Universidad de Córdoba, 14014 Córdoba, Spain; 4Hospital de Referencia La Equina, 29691 Manilva, Spain; juan@laequina.com; 5Clinique Vétérinaire Équine de Grosbois Altano-Gruppe, 94470 Boissy-St-Léger, France; genton.martin@gmail.com (M.G.); fabricerossignol94@orange.fr (F.R.); 6Méndez Hospital Equino, 14100 La Carlota, Spain; mendezhospitalequino@gmail.com; 7Unitat Equina de l’Hospital Clínic Veterinari, Universitat Autònoma de Barcelona, 08193 Barcelona, Spain; frederic.climent@uab.cat; 8Departament de Medicina i Cirurgia Animals, Facultat de Veterinaria, Universitat Autònoma de Barcelona, 08036 Barcelona, Spain; marta.prades@uab.cat; 9Independent Researcher, 08551 Tona, Spain; immaroquet@hotmail.com; 10Hospital Clínico Veterinario, Departamento de Medicina Animal, Universidad de Extremadura, 10003 Cáceres, Spain or manuel.iglesias@universidadeuropea.es (M.I.); ezquerra@unex.es (J.E.); 11Hospital Clínico Veterinario, Departamento de Medicina y Cirugía Animal, Universidad CEU-Cardenal Herrera, CEU Universities, 46115 Alfara del Patriarca, Spain; ana.vellosoalvarez@uchceu.es; 12Departamento de Medicina y Cirugía Animal, Facultad de Veterinaria, Universidad Complutense de Madrid, 28040 Madrid, Spain

**Keywords:** horse, laparoscopy, inguinal hernioplasty, barbed suture

## Abstract

**Simple Summary:**

Inguinal hernias (IHs) occur uni- or bilaterally when intestinal loops enter the vaginal ring (VR), exiting the abdominal cavity. They are a relatively common problem in foals and mature horses, mostly affecting intact male animals. In stallions (acquired hernias), the clinical scenario presents with colic and is an emergency, life-threatening condition, while in foals (congenital hernias), conservative management could be carried out. However, surgical treatment may be necessary as a preventive or therapeutic measure to close the VR. Several laparoscopic techniques have been reported, each with different results and postoperative complications. This retrospective study describes the results using barbed sutures. Barbed sutures are characterized by the presence of “barbs” along their length, which provide better tissue engagement and eliminate the need for knots, which is a great advantage in minimally invasive surgery. This technique was applied to 40 animals, successfully closing a total of 59 VRs. Barbed sutures were used alone or in combination with other techniques, and postoperative follow-up data was gathered from owners and/or referring veterinarians. This multicentric retrospective study suggests that laparoscopic hernioplasty using barbed sutures is a safe and effective method for the treatment and prevention of inguinal hernias in horses.

**Abstract:**

Inguinal hernias (IHs) and ruptures are a relatively common condition in horses, occurring in foals (congenital) and adult (acquired) animals. A retrospective observational analysis was conducted on 40 cases that underwent laparoscopic surgery to close the VRs using barbed sutures alone or combined with other techniques. Signalment, clinical presentation, surgery, and follow-up data were obtained. In total, fifty-nine VRs were closed using barbed sutures (alone or in combination with other methods), with six cases performed prophylactically and forty-four due to acquired IH. Of the forty-four cases with IH, four were non-strangulated hernias, while thirty presented with strangulated small intestines (twenty-eight acquired and two congenital). The results obtained in this study suggest that laparoscopic hernioplasty with barbed sutures is an effective and safe surgical procedure that could be recommended as a standard practice for managing inguinal hernias in horses, particularly when sparing testicles or preserving reproductive capabilities is a priority.

## 1. Introduction

Inguinal hernias and ruptures are a relatively common condition in horses, particularly in intact males. This condition is characterized by the protrusion of abdominal contents, typically small intestine, through the vaginal ring (VR). Herniation can be classified as either congenital or acquired. It occurs when the protrusion occurs through the VR into the vaginal tunic and, in intact males, in the scrotal sac. A hernia is considered ruptured when protruded intestine through the VR lies subcutaneously due to a tear in the parietal layer of the vaginal tunic [1]. Most equine hernias are indirect, with the intestine herniating through the VR. Direct inguinal hernias, caused by a rupture of the abdominal wall tissues in the inguinal area, are extremely rare in horses, unlike in human medicine [1]. Acquired strangulating inguinal herniation is a potentially life-threatening condition in stallions. Several studies have investigated the clinical presentation, diagnosis, and treatment of this condition in horses [2,3,4,5,6]. It usually presents with signs of acute and severe abdominal discomfort, and oftentimes, a firm, cold, and enlarged affected testicle can be palpated [7]. While acquired inguinal hernias are generally present on one side, surgical treatment often involves preventive bilateral castration or excision of only the affected testicle. This is followed by reducing the herniated contents, possibly with concurrent intestinal resection of the affected bowel, and repairing the hernia defect by suturing the external inguinal ring to prevent recurrence. In some cases, the hernia can be reduced by an external massage under general anesthesia and may resolve without the need to castrate the affected testicle [8].

When the owner prefers to preserve the stallion’s testicles and due to the life-threatening and emergency characteristics of strangulated herniation, prophylactic laparoscopic closure of the VR can be performed in the same or the contralateral VR. This procedure closes the VR partially.

Congenital inguinal herniation in foals is diagnosed at birth by noticeable swelling in the inguinal region, filled with intestinal contents. This condition rarely leads to strangulation obstruction due to the large diameter of the inguinal rings. Diagnosis is typically confirmed through physical examination and ultrasonography. This condition is often congenital, and the recommended management includes daily manual reduction and bandaging before considering surgical intervention. If these approaches fail, if the hernia expands, or if the vaginal tunic ruptures, further surgical interventions are required [9].

In scenarios where preserving the stallion’s reproductive capabilities is preferred, laparoscopic closure of the VR may be performed as a preventive measure, either on the affected side or contralaterally, partially sealing the VR. Laparoscopic surgery, favored in recent years for its numerous advantages, such as quicker recovery, reduced pain, and fewer complications compared with traditional open surgery, has led to the development of various testicle-preserving laparoscopic hernioplasty techniques. These include intracorporeal suture closure of the VR [9,10], cylindrical polypropylene mesh prosthesis [11], peritoneal flap hernioplasty techniques [12,13], cyanoacrylate [14], a tacked intraperitoneal slitted mesh technique [15] or using a surgical anchoring system [16].

Barbed sutures are a unique type of sutures featuring small projections or “barbs” along their length. These barbs enhance tissue grip and eliminate the need for knots to secure them in place. Several studies have reported the use of barbed sutures in equine surgery in both open [17] and laparoscopic procedures [10,18,19,20,21]. Barbed sutures (mono-or bidirectional) avoid the need for intracorporeal knots [18]. In vitro and ex vivo studies in horses have demonstrated advantages such as a reduction in exposed suture material due to knot absence in a jejunum end-to-end anastomosis model [22], equivalent bursting strength to standard sutures in a cadaveric bladder equine model [23], and faster results with less suture material left in the tissue and sustained comparable bursting pressure when compared with unbarbed sutures in a pelvic flexure enterotomy model [24].

Barbed sutures might also be advantageous for laparoscopic VR hernioplasty due to their ease of use and the avoidance of demanding laparoscopic procedures or the use of more expensive consumables. Nevertheless, to the authors’ knowledge, their use has only been primarily documented in foals [25,26]. This study aims to assess the safety and effectiveness of a laparoscopic hernioplasty technique using knotless barbed sutures to close the VR in several clinical conditions, in foals and adult horses, and to provide follow-up data on the outcome.

## 2. Materials and Methods

A retrospective observational analysis was conducted on cases collected from November 2017 to January 2024, involving eight hospitals and three freelance equine surgeons. Inclusion criteria were laparoscopic hernioplasty performed using barbed sutures at least in one VR. Cases included animals of any age, encompassing both foals and adults, and involved laparoscopic closure of one or both VRs using barbed sutures solely or in conjunction with other techniques. The study included cases of post-inguinal hernia hernioplasties as well as prophylactic surgeries in horses without a prior episode of herniation.

For animals undergoing prophylactic hernioplasty, a detailed anamnesis was taken to ascertain the rationale behind the preventive surgery. In cases with a prior inguinal hernia, collected data included the hernia type, categorized as strangulated (presenting acute colic symptoms) or non-strangulated (lacking acute colic symptomatology) and whether it was unilateral or bilateral, specifying the herniated side. The method of reduction, including external massage, abdominal laparotomy, inguinal/scrotal approach, or reduction during laparoscopy, was noted. Additionally, data on the necessity for intestinal resection, anastomosis, and castration on the affected side were included. The interval between the inguinal hernia episode and the laparoscopic hernioplasty was also recorded.

For both situations, the following data were recorded: the duration of pre-operative fasting, whether the procedure was standing laparoscopy or under general anesthesia, and the number and location of laparoscopic portals. The extent of the procedure was detailed, including whether closure involved only the herniated side’s VR, the contralateral side, or both. The technique employed for each VR was described (manual suture with laparoscopic needle-holder or with Endo Stitch™ or SILS Stitch™, combined with V-Loc™ suture), identifying whether barbed suture was used alone or in combination with other methods. Barbed suture type and specific sutured areas of the VR were also documented. Finally, records were kept on perioperative and postoperative medication, intraoperative incidents, hospitalization duration, and complications.

Follow-up data were obtained from owners and/or referring veterinarians, inquiring about postoperative incidences, particularly the external aspect of the testicles and any instances of post-hernioplasty herniation or recurrence, specifying if it occurred on the same or the contralateral side. For breeding stallions, additional reproductive monitoring data, including alterations in sexual behavior and fertility or spermiograms (if available), were collected.

Animal care and use committee approval was not necessary for this retrospective study with medical records. This is confirmed by the Ethic Committee for Animal Experiments from the University of Zaragoza (project number PI20/23NE).

Descriptive statistics were employed to summarize these data. For continuous variables, the maximum and minimum values, means ± standard deviation (or medians and interquartile range -IQR- for non-normally distributed data) were calculated using statistical software (GraphPad Prism 10.2.1). Categorical variables were summarized using frequencies and percentages.

## 3. Results

### 3.1. Signalement and Clinical Presentation

Records of 40 horses that underwent VR laparoscopic hernioplasty using barbed sutures were included in this study. Six underwent prophylactic surgery, and thirty-four were operated on following signs of inguinal hernia (three congenital and thirty-one acquired). The mean age of the 40 animals was 5.9 ± 3.4 years; the age ranged from 57 to 91 days in three congenital cases, from 3 to 6 years in prophylactic cases, and from 7 months to 16 years in acquired inguinal hernia cases. Most (67.5%) of the horses were Pure Spanish Breed (PRE). This breed was also predominantly represented in congenital cases (2 of 3) and in prophylactic cases (5 of 6). The other breeds were 3 French Warmblood, 2 KWPN, 1 Anglo–Arabian, 1 Arabian, 1 French Trotter, 1 French Pony, 1 Hanoverian, 1 Oldenburg, and 1 Spanish Warmblood. In eight cases, body weight was not registered. In the rest, the median weight was 500 kg, and the IQR varied between 455 and 546 (range in three congenital cases: 100–150 and 250–680 in the other thirty-four acquired or prophylactic cases).

Among the six prophylactic cases with no previous signs of inguinal hernia, the primary reason for surgery was the owner’s intent to prevent a possible hernia episode (four cases, all PRE), one French Warmblood with a family history of inguinal hernia, and one PRE with bilateral hydrocele. In one of these cases, small bowel herniation on the left side was detected during laparoscopy without the animal showing any signs of colic or inguinal herniation.

Of the three cases with congenital hernia, one of them (with bilateral hernia) underwent preventive surgery without having shown previous signs of colic or strangulated hernia. The other two foals, after weeks of conservative treatment, showed signs of colic due to inguinal herniation. The second foal was operated at 3 months of age due to a bilateral inguinal hernia; it was resolved by external massage without surgery, and laparoscopic bilateral hernioplasty with testicular sparing was performed 6 days later. The third foal (Figure 1) was born with a right-sided hernia. Treatment was attempted with subsequent manual reductions, but at the age of two months, he showed signs of acute colic and was referred for emergency surgery. During laparoscopy, a herniated ileum through the right VR was observed. After hernia reduction, combining traction with atraumatic laparoscopic forceps and external scrotal massage, hernioplasty was performed on the herniated side only.

Of the 31 cases of acquired inguinal hernia, one was bilateral, 16 were on the left side, and 14 on the right. Three out of thirty-one (9.7%) presented with non-strangulated inguinal hernias without colic symptoms. One of them was a 12-year-old PRE horse, which the referring veterinarian had diagnosed with recurrent inguinal hernia on the right side based on clinical and ultrasonographic evidence. The horse was referred for bilateral orchidectomy with a primary closure technique. During surgery in dorsal recumbency, although not herniated, the intestine was observed, and the right inguinal canal was found to be significantly enlarged, both on rectal palpation (VR) and by palpation during surgery (external inguinal ring). Consequently, after castration and suturing of the external inguinal rings, it was decided to perform a laparoscopic exploration, and a right-sided hernioplasty was performed. The second case involved an Anglo–Arabian, aged 2 years and 8 months, who was castrated at 10 months of age. Fourteen months later, when the horse began its training, following a period in the pasture after castration, clinical signs of intermittent right-sided inguinal hernia were observed and confirmed by ultrasound. The animal was referred for inguinal hernioplasty on that side. The third of these cases was a horse with a chronic, non-strangulated left inguinal hernia. Six weeks previously, the horse had undergone bilateral laparoscopic hernioplasty by an external surgeon. The left-sided hernia recurred 4 days after that first operation, and 6 weeks later, the horse underwent repeated hernioplasty but only of the re-herniated left VR.

A summary of the distribution of cases is given in Figure 2.

In 30 cases involving acquired (28 horses) or congenital (2 foals) hernias in horses experiencing colic symptoms attributed to inguinal hernias, the average time between colic onset and hernioplasty procedure was found to be 22 days, with a range of 0 to 1298 days and an IQR of 11 to 82 days. In terms of treatment, 23 cases underwent attempted reduction by external massage, succeeding in 15 cases (65.2%). Of these cases, 16 underwent ventral midline laparotomy, while 12 cases received parainguinal, inguinal, or scrotal incisions, often combined with ventral laparotomy. During the episode of colic due to inguinal hernia, the testicle on the affected side was castrated in 10 of the 30 cases (33.3%), all of which had unilateral hernia.

### 3.2. Overview of Surgical Procedures

Laparoscopic hernioplasty was performed uni- or bilaterally in 40 animals:2 horses were previously castrated: 3VRs were sutured, one bilateral and one unilateral.6 animals underwent bilateral hernioplasty with testicle preservation prophylactically.32 intact males with previous inguinal hernia
○In twenty-two horses in which both testicles were preserved after an inguinal hernia episode, three had bilateral and nineteen unilateral hernias (in two of them, only the affected side underwent hernioplasty).○Ten cases were previously castrated of the affected testicle during the episode of colic due to an inguinal hernia, so hernioplasty could only be performed in ten contralateral VRs (with no previous inguinal hernia on that side). Cases where the affected testicle was castrated during the episode of colic due to inguinal hernia, were also included, as hernioplasty of the contralateral testicle on that side is performed to avoid possible herniation on the opposite side.

Therefore, 67 VR hernioplasties were performed in these 40 cases. Eight of them (4 previously herniated, 3 prophylactic, and 1 without testicle) were performed without using barbed sutures. These eight horses have been included in the study because the contralateral VR hernioplasty was sutured with barbed suture (alone or in combination with other methods).

In consequence, hernioplasty with barbed suture was performed only on 59 VRs (Table 1):56 sparing testicles ○22 on the previously herniated side○25 on the contralateral (without previous inguinal hernia on the operated side)○9 prophylactic (without a history of inguinal hernia on any side)3 without testicles

These 59 hernioplasties include VRs in which the barbed suture was used alone or in combination with other methods according to this distribution:46 with barbed sutures only (17 in previously herniated VRs, 19 in contralateral, 7 prophylactic, and 3 without testicles).13 combining barbed sutures and other additional methods (5 in previously herniated VRs, 6 in contralateral, and 2 prophylactic).

### 3.3. Surgical Technique with Barbed Suture

#### 3.3.1. Preparation for Surgery and Perioperative Medication

In one 3-month-old foal undergoing emergency surgery for colic due to an inguinal hernia, fasting was not an option. In another lactating foal, fasting lasted 6 h. In the remaining 38 cases, the median pre-surgical fasting time was 24 h (range 12 to 72, and IQR 24 to 48 h).

In 35 adult animals, the surgery was performed standing with horses restrained in stocks, with different sedation protocols and locoregional anesthesia. The other five cases were operated on under general inhalation anesthesia in the Trendelenburg position. This group included four foals less than 7 months old and one adult horse in which anesthesia was also used to treat an incisional hernia that had appeared after colic surgery for an inguinal hernia 6 months earlier.

Horses received perioperative antibiotic coverage with a median duration of 2.5 days (range 2 to 7, IQR 2 to 3 days). A combination of penicillin and gentamicin was administered to 18 horses, and only 21 horses received penicillin. One foal that presented signs of colic due to an inguinal hernia received clarithromycin and cefquinome because a rhodococcal infection was diagnosed. Only one case did not receive anti-inflammatory medication. In the rest, anti-inflammatory drugs were used for a median of 5 days (range 2 to 7, IQR from 2 to 3 days). Flunixin meglumine was used in 22 horses, and phenylbutazone in 17. One horse started with flunixin and then switched to phenylbutazone.

#### 3.3.2. Surgery

In the five cases where surgeries were performed using general anesthesia, the laparoscope port was positioned either at the umbilicus or in its proximity. Furthermore, in two cases (out of three) or three cases (in one bilateral procedure), supplementary portals were utilized for the surgical instruments. In one of these five cases, the location of the portals was not recorded. Thirty-five adult horses underwent standing laparoscopic surgery, and in four cases, the location of laparoscopic portals was not specified. In twenty of thirty-one animals (64.5%), three portals were used in the operated flank (in five of them, one of these portals was used between the last two ribs). In eleven cases (35.5%), four portals were placed per flank (seven of them with one portal between the last two ribs). In most cases, three portals are used: one for the optic, one for the automatic suture device, and one for an additional instrument to assist in tissue manipulation or positioning for suture application. In the cases where four portals were used, it was normal to have another instrument helping to push back the testicular cord when it was necessary to use a viscera retractor to move intestinal loops that interfered with the field of view and in the case where manual barbed suturing was used, without automatic suture device. The approximate location of these portals was: one in the paralumbar fossa, 15 cm ventral to the transverse processes of the lumbar vertebrae, midway between the last rib and the tuber coxae, one portal 7 cm ventral and 1–2 cm caudal to the previous portal, one portal 7 cm ventral and 1–2 cm caudal to the previous portal and another portal at the level of the tuber coxae ventral and between the last two ribs.

In most cases (92.5%), an automated laparoscopic suturing device (thirty-three using Endo Stitch™ and five using SILS Stitch ™, Medtronic, Dublin, Ireland) was used to suture the VR, in combination with commercial custom-made reload of unidirectional barbed sutures V-Loc ™. These sutures are threaded with a needle that allows them to be used with the Endo Stitch or SILS Stitch system and have a loop at the end, which avoids tying an initial knot. Only in two cases was a laparoscopic needle holder used for suturing, using sutures: Filbloc 36 mm 1/2 circle, USP 1, 15 cm, absorbable (polydioxanone), unidirectional, with a stopper at one of the suture line (final lock system) (Ref. FQ24EHHAB) and 2/0 USP trimethylene carbonate glycolide monofilament bearded absorbable suture with CT-2 needle. In 36 of the 40 cases (90.0%), absorbable sutures of a copolymer of glycolic acid and trimethylene carbonate (V-Loc 180 series, 15 to 20 cm in length) were used. In 27 (67.5%) cases, 2/0 USP sutures were used, and in 12 (30.0%), 0 USP sutures were used.

In general, a technique similar to that described in previous studies on VR closure in foals was used [25,26]. Barbed suture (alone or in combination with other techniques) was used in 59 VRs. Three of them were VRs in which the testicle had been previously castrated. In 42 of the remaining 56 VRs (75.0%), only the craniolateral aspect of the VR was sutured (Figure 3). In 14 VRs (25.0%), both the craniolateral aspect and the caudomedial side of the testicular cord were sutured (Figure 4).

In 10 cases, combining barbed sutures and other additional methods was used for the hernioplasty of one or both VRs. In eight of these horses, both VRs were closed, and in two, the hernioplasty was unilateral (the contralateral testicle had previously been castrated, and the external inguinal ring had been closed), for a total of 10 horses with 18 VRs with hernioplasty. In 13 of these VRs, barbed sutures were used in combination with other methods. In most of these VRs (10), barbed sutures and cyanoacrylate were used. The glue was used to reinforce the hernioplasty, and the cyanoacrylate was applied after suturing the VR using the technique previously described [14]. The other three hernioplasties were two in a foal with a bilateral hernia in which both VR were treated with barbed sutures and, in addition, a single suture stitch with a non-barbed suture in each VR. The other was in a horse that had been previously castrated, but due to the large size of the inguinal canal, it was decided to perform a laparoscopic post-castration hernioplasty of the VR in addition to suturing the external inguinal ring with non-barbed sutures. In this case, in addition to suturing the VR with a barbed suture, part of the mesorchial fold was included in the VR suture. As this is an additional variation to the simple suturing of the RV with barbed suture, this case has been included in this section.

#### 3.3.3. Postoperative Care

The median duration of hospitalization after hernioplasty was 5 days (range 2–18, IQR 3 to 6.7 days). The animals received postoperative antibiotics and anti-inflammatory medication, as indicated in Section 3.3.1.

Following surgery, in most horses, two weeks of regulated physical activity was prescribed for each horse, involving exclusively 10–20 min of hand walking per day. Subsequently, a gradual return to regular exercise levels was initiated, during which the majority of the horses were able to resume exercise at the same intensity as before the surgical intervention after a few weeks.

### 3.4. Complications and Outcome

#### 3.4.1. Intraoperative Incidences and Complications

During hernioplasty, herniated contents were found in six cases. Only one of these animals showed signs of colic: a 3-month-old foal with a congenital hernia that underwent emergency surgery due to acute colic symptoms; during the same laparoscopy, the ileal hernia was reduced, and the hernioplasty was performed. The other five cases were two foals with congenital non-strangulated small bowel hernia (without colic) in which the hernia was reduced at the same surgery before hernioplasty, two adult horses that previously had colic due to inguinal hernia and during laparoscopy for hernioplasty an omental hernia with adhesions to the testicular cord was detected, and omentectomy had to be performed before hernioplasty and, finally, one horse undergoing prophylactic hernioplasty (with no previous history or signs of herniation) in which there was an incidental finding of small bowel hernia, which was reduced prior to hernioplasty. In these cases, the herniated bowel was reduced by traction with atraumatic laparoscopic forceps; however, in the two foals with hernia without colic, the hernia was reduced spontaneously by the effect of the Trendelenburg position with the capnoperitoneum, and it was even necessary either to have an operator holding the testicles in the scrotal sac from the outside or to make an inguinal incision to hold the testicle extra-abdominal with forceps.

In two animals, minor bleeding or a small hematoma of about 4 cm occurred when the barbed suture was being sutured, and the testicular cord was punctured (one case) or by puncturing the external pudendal or epigastric vessels adjacent to the caudomedial part of the VR (in the other case). Both incidents were resolved without consequences and the need for further intervention.

#### 3.4.2. Early Postoperative Period

During this period, incidences and complications were recorded in seven animals: two cases with moderate emphysema around some laparoscopic portals, one horse with hematoma in the transverse facial artery, one animal with mild dorsal patellar fixation occurring 4 days after surgery (adult operated under general anesthesia), one reoperated case that maintained moderate inflammation of the left hemiscrotum at 14 days but without pain on palpation, one horse with colonic impaction 7 days after surgery, and one case in which tachycardia and hyperthermia with profuse diarrhea were observed after surgery. All of them were resolved satisfactorily. The remaining 33 cases (82.5%) did not have any complications during hospitalization or the early postoperative period.

#### 3.4.3. Inguinal Hernia Follow-Up and Outcome

The median follow-up time was 15 months (range 0–60, IQR 12 to 30 months). In 31 cases, the follow-up is one year or longer. Of the nine cases with a follow-up of less than one year, four could not be followed or were sold, and in five cases, 12 months had not elapsed between the date of hernioplasty and the writing of the manuscript.

Of the forty horses, two cases of inguinal herniation after hernioplasty (5%) were documented among the forty cases during the follow-up period. Focusing exclusively on thirty-one cases with follow-up of one year or more, five were preventive, and twenty-six had a history of previous inguinal hernia (twenty-four acquired and two congenital), of which twenty-two had signs of colic due to hernia. In these 31 cases, hernioplasty was performed in 51 VRs, 21 of which were VRs on the previously herniated side and 30 were not (20 contralateral to the herniated side and 10 VRs from cases with prophylactic closure). Among them, herniation after hernioplasty was recorded in two VRs (6.5%), one of them in a previously herniated VR (1 of 21, 4.8%) and another in a contralateral VR (1 of 20, 5.0%). Among the five preventative cases (10 VRs), no herniation was recorded post hernioplasty. Table 1 and Figure 2 indicate in brackets the incidence of post-hernioplasty hernia according to the type of case and the type of technique used.

One of these two animals with inguinal hernia post-hernioplasty was a 6-year-old PRE horse that had colic due to a left inguinal hernia that could be reduced by external massage. Hernioplasty was performed, sparing the testicles on both sides, using only barbed sutures on both sides, placed with an Endo Stitch™. The horse re-herniated 2 months later, again on the same left side. The other was an 8-year-old SF stallion who had colic due to a left inguinal hernia, and the testicle was removed. The contralateral (right) side underwent hernioplasty, sparing the testicle 3 years and 7 months later. The hernioplasty was performed using a combination of barbed sutures placed with Endo Stitch™ and Surgibond. A total of 2 years and 10 months after hernioplasty, the herniation occurred on this right side, contralateral to the initially herniated side. In both cases, only the craniolateral part of the VR was sutured.

#### 3.4.4. Reproductive Follow-Up and Outcome

Of the 34 males 3 years old and older, eighteen horses that owners wanted to keep intact did not carry out breeding activities, fifteen were used as breeding stallions, and one was lost to follow-up.

Among these fifteen stallions, five have not covered mares again, one because of a re-herniation in the same side after 2 months, and this testicle was castrated without having restarted his reproductive activity, and four because the hernioplasty was still recent. In the remaining 10, no adverse effects on fertility have been reported, with offspring or pregnancies after hernioplasty (nine of them with a follow-up of more than one year). However, there is no information on comparative fertility analysis with respect to the situation prior to hernioplasty for any of them. In three of these stallions, spermiograms before and after hernioplasty were available, and no remarkable alterations were observed.

## 4. Discussion

To the author’s knowledge, this is the first retrospective multicentre study describing the use of barbed sutures to close the VR in a larger population of adult horses with acquired inguinal hernias, including surgical technique, perioperative complications and data on both short and long-term outcomes. Collecting this information is an important step in assessing the actual advantages of this type of suture and the outcome obtained. As mentioned, several laparoscopy herniorrhaphy techniques have been described in horses [7,10,11,13,14,15,16,27,28]. In general, the results obtained with the techniques have been satisfactory. However, the difficulties in performing this surgery and additional costs related to the type of implants used to close the VR could be limiting aspects of several laparoscopy techniques when compared with the use of barbed sutures [7,10,13,14,15,27,28]. Considering that the automatic suturing devices, although designed for single use in human medicine, can be gas sterilized and reused many times, the cost of consumables for this technique with barbed sutures is less than that required for other techniques using expensive mesh and laparoscopic staples [15].

The first report of a laparoscopic herniorrhaphy with barbed sutures was described in an adult gelding horse [25] and then in nine foals with congenital inguinal hernias [26]. In the present case series, we present the results of 30 adult horses that had an acquired inguinal hernia operated mainly in a standing position (29 horses) and others under general anesthesia. This study aimed to describe and amplify the veterinary literature by presenting a wide range of clinical cases treated by using barbed sutures, including prophylactic, congenital, and acquired inguinal hernias in several clinical conditions, such as incarcerated and strangulated hernias, in previously herniated VR or contralateral VR (herniated or not).

When the horses of our study were operated in a standing position, three to four portals were employed instead of two, as previously described for the VR closure of an adult horse [25]. We believe that the main advantage of an extra portal is the possibility of inserting atraumatic forceps to manipulate the testicular cord in order to avoid testicular cord trauma and to be able to close the craniolateral and even the caudomedial portion of the VR. In general, most of the cases performed under general anesthesia were operated with a technique similar to that described by Maurer et al. [26].

Barbed sutures represent an innovative advancement in the development of surgical material, offering several advantages over traditional sutures. These sutures are characterized by having small projections or “barbs” along their length, improving tissue retention and eliminating the need for knots. The barbs allow for good tissue engagement, leading to better wound closure and reduced risk of suture loosening or slippage. Barbed sutures also eliminate the need for tying knots, streamlining the suturing process and reducing surgical time. This is particularly advantageous in laparoscopic procedures using intra-corporeal suture techniques where efficiency and precision are crucial [29,30].

Several studies have demonstrated that the use of barbed sutures in equine surgery provides equivalent or even superior strength compared with traditional sutures in various equine surgical models, ensuring reliable wound closure and tissue approximation [31,32,33,34,35]. Furthermore, barbed sutures result in less suture material remaining in the tissue compared with traditional sutures, potentially contributing to faster healing and reduced risk of complications such as suture abscesses or tissue irritation. However, their use requires surgeons to adapt their technique and familiarize themselves with the unique characteristics of these sutures [31,32,33,34,35]. Proper training and practice are essential to minimize the risk of complications; despite their advantages, barbed sutures may pose challenges, such as the potential for suture entanglement or difficulty in adjusting tension once the suture is placed. Surgeons must exercise caution and skills when using these sutures to mitigate such risks [36,37,38,39].

Intraoperative complications were observed in two animals from our case series. These complications were transient bleeding or testicular cord hematoma formation. In general, these results are in concordance with those observed by Maurer et al. [26]. These types of complications could be related to an accidental puncture of a major blood vessel either during abdominal trocar allocation, which is inherent to any laparoscopic technique [40], or by puncturing the testicular vascular package during the VR closure but did not require any further management measure in these two animals.

In this study, three foals with congenital hernias were included. These animals did not present any complications during surgery or follow-up, with similar results to those observed by Maurer et al. [26]. We also described the use of the technique for the prevention of inguinal hernias in six adult horses, in which the testes were spared. At this point, it is necessary to consider that only 2/6 horses were intended for reproduction. In the rest of the horses, mainly PRE horses, the testes were spared at the owner’s request as it is very common to keep the males intact in this breed due to good temperament and for cultural and commercial reasons.

In total, 10 breeding stallions that continued to breed after hernioplasty surgery have been included in this study. In all these stallions, offspring or pregnancies have been reported after hernioplasty (nine of them with a follow-up of more than one year). The potential impact of hernioplasty on the reproductive functionality of stallions is an important aspect of assessing the safety of these techniques. However, this aspect has been studied in more detail only in experimental animal studies that analyzed the impact of some laparoscopic hernioplasty techniques on spermiograms [16,41] and testicular perfusion [16,27]. One of the limitations of the present study is that this could only be assessed in the three stallions indicated in 3.4.4. for which pre- and post-hernioplasty spermiograms are available. Nevertheless, these assessments have also not been carried out in any of the published case studies and laparoscopic hernioplasties. Generally, as in our study, these studies do not provide concrete data on fertility but only mention the preservation of reproductive function or the absence of negative impact on it [12,13,14,15].

In the present study, the vast majority of the hernioplasties with barbed sutures were performed using two different suturing devices, which included the Endo Stitch™ device (Medtronics) in thirty-three cases (95%) and SILS Stitch ™ device (Medtronics) in seven cases. Notably, the SILS Stitch ™ device is based on the same technology as the Endo Stitch™ suturing device, allowing better articulation of movements (up to 75°) with a rotation of up to 360°. These features allow the surgeon to work in tight spaces and to reach tissues in their anatomical position, avoiding pulling or stressing them. However, the SILS Stitch™ device is more expensive than the Endo Stitch™. In general, the use of both suturing devices allows for the effective anchoring of the barbed suture, and although the time of suturing was not presented and compared with other suture management techniques in this study, all the surgeons that participated described that the automatic device significantly simplifies surgical procedures.

Recurrence has been described mainly using the peritoneal flap technique. In this study, 2/40 horses (5% recurrence rate) developed recurrence of hernia, while in horses in which the peritoneal flap was performed, 4/30 cases (13% recurrence rate) were described [13]. Interestingly, horses that presented recurrence in the mentioned study had an incomplete closure. No cases of recurrence were documented with other published surgical techniques used for laparoscopic VR closure in horses, except for the peritoneal flap approach mentioned above. In the horses that re-herniated in this study, only the craniolateral portion of the VR was closed. Both results suggest that closure of the craniolateral and medial portion of the VR might decrease the risk of recurrence and that closure of the VR, specifically addressing both the craniolateral and caudomedial portions, could be pivotal in laparoscopic inguinal hernioplasty to mitigate the risk of hernia recurrence effectively, which has been previously pointed out by other authors [13,14,16]. Suturing the caudomedial part can be difficult and must be performed with caution due to the possibility of puncturing the epigastric vessels, which originate from the epigastric pudendal trunk [42]. This scenario occurred in one of the cases in this study, but it was resolved without complications. To avoid suturing this area and prevent re-herniation in the caudomedial region, it can be very useful to push back the testicular cord while suturing only the craniolateral part.

This study presents certain limitations. The main one was the absence of a control group, which is the most common pitfall in studies of the same nature in horses where other laparoscopic techniques were described [7,10,13,14,15,27,28]. Even though this case series includes a large variety of cases (congenital and acquired hernias or adult and foals), which could have been considered a potential limitation, it emphasizes the versatility of this procedure to produce successful outcomes in different scenarios. Surgical time was not recorded since the objectives were more about the outcome and benefits of the technique. To guarantee VR closure, a second exploratory laparoscopy should have been strongly recommended at 2–3 months, but this was not possible due to the clinical nature of the study and the owner’s and/or financial constraints.

## 5. Conclusions

This multicentric retrospective study provides compelling evidence that laparoscopic hernioplasty using barbed sutures is both a safe and effective method for treating and preventing inguinal hernias in horses. Importantly, this technique has shown no detrimental impact on offspring ability, making it particularly suitable for use in breeding stallions. However, further studies specifically designed to evaluate the reproductive and fecundity effects of stallions treated with this technique in a larger number of cases would be necessary. Notably, compared with the other laparoscopic methods, this technique is less technically demanding, especially when automatic suturing devices are used. Overall, the positive outcomes in this study support that laparoscopic hernioplasty with barbed sutures could be recommended as a standard practice for managing inguinal hernias in horses, especially when the goal is to preserve testicles and reproductive capabilities.

## Figures and Tables

**Figure 1 animals-14-01826-f001:**
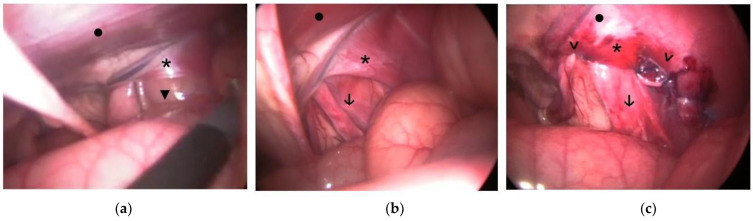
Images of the laparoscopic hernioplasty of a 6-month-old foal with congenital right inguinal hernia admitted to undergo emergency surgery under general anesthesia due to acute signs of colic: (**a**) Indirect hernia of the ileum. (**b**) Appearance of the VR after reduction in the hernia by traction with atraumatic laparoscopic forceps. (**c**) Image after hernioplasty using only barbed sutures. ▼: incarcerated ileum, *****: vaginal ring, •: abdominal wall, **↓**: testicular cord, **>**: barbed sutures.

**Figure 2 animals-14-01826-f002:**
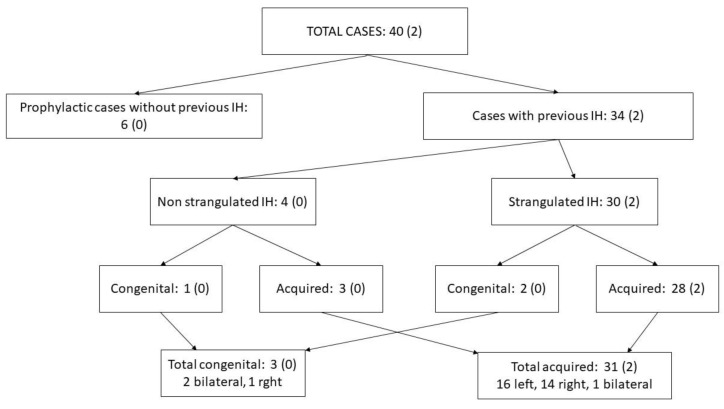
Distribution of horses included according to case type. The amount of post-hernioplasty herniations is recorded in brackets. IH: inguinal hernia.

**Figure 3 animals-14-01826-f003:**
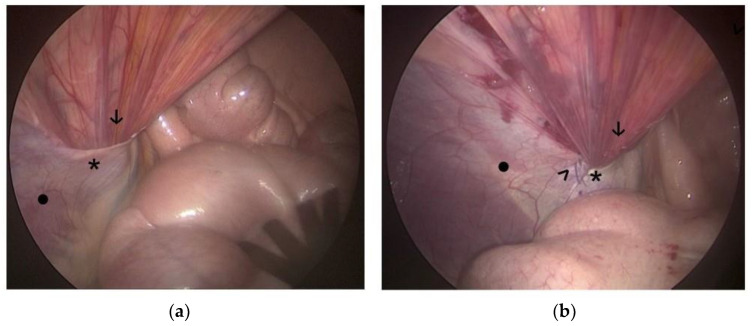
Images of laparoscopic hernioplasty of a right VR. (**a**) Appearance of the VR before hernioplasty; (**b**) Image after hernioplasty using only barbed sutures, suturing only the craniolateral aspect of the VR. *****: vaginal ring, •: abdominal wall, **↓**: testicular cord, **>**: barbed sutures.

**Figure 4 animals-14-01826-f004:**
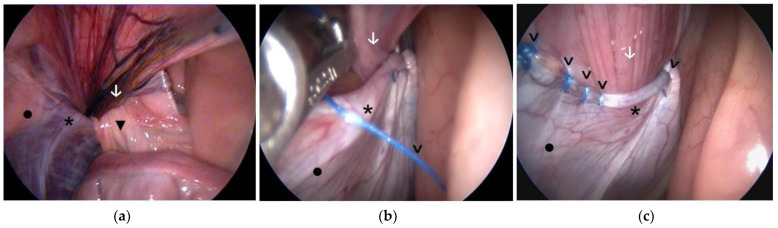
Images of the laparoscopic hernioplasty after acquired inguinal hernia. (**a**): View at the beginning of the surgery, with herniated omentum; (**b**) Closure of the caudomedial part with barbed suture after reducing the omentum herniation by traction with laparoscopic forceps; (**c**) Image after hernioplasty using only barbed sutures, suturing both the craniolateral and caudomedial parts of the VR. ▼: herniated omentum, *****: vaginal ring, •: abdominal wall, **↓**: testicular cord, **>**: barbed sutures.

**Table 1 animals-14-01826-t001:** Distribution of vaginal rings (VR) in which hernioplasty with barbed suture was performed, stratifying by the type of technique used and whether it was a previously herniated VR, its contralateral side, prophylactic technique in a horse without previous hernia, or complete closure in a VR without testicle. The number of re-herniations was recorded in brackets.

Hernioplasty Technique	Sparing Testicle			Without Testicle	Total
	Previously Herniated	Contralateral (WPH)	Prophylactic (WPH)		
**Only BS**	17 [1]	19 [0]	7 [0]	3 [0]	46 [1]
**BS + other methods**	5 [0]	6 [1]	2 [0]	0 [0]	13 [1]
**Total**	22 [1]	25 [1]	9 [0]	3 [0]	59 [2]

BS: barbed sutures. WPH: without previous hernia in this VR.

## Data Availability

The original contributions presented in the study are included in the article; further inquiries can be directed to the corresponding author.

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
