# Peer review of "Use of Knotless Barbed Sutures in Laparoscopic Inguinal Hernioplasty in Horses: 40 Cases"

_animals, 2024, doi:10.3390/ani14121826_

Round 1

Reviewer 1 Report

Comments and Suggestions for Authors

General concepts:

I think this paper has information in it that deserves publication, however the overall impression of the paper is very convoluted and hard to track the details. I think there are ways that the presentation of the data could be improved and more clearly described.

I have a major problem with the case inclusion – the end of the introduction says “This study aims to assess the safety and effectiveness of a laparoscopic hernioplasty, with a focus on preserving the testicle, using knotless barbed sutures to close the VR in foals and adult horses and to provide a follow-up data on the outcome” and cases that were castrated and barbed suture were not used were included. I think it would be OK to say 40 horses and 67 vaginal rings were included, of these 13 cases were castrated and 8 vaginal rings were closed without barbed suture – these cases were excluded, leaving X number of horses in the study.

Also 17 authors seems like a lot, if some are purely providing cases, that doesn’t automatically deserve authorship. From the author contribution statement, it does not seem that all authors qualify for the 4 criteria required by the ICMJE guidelines that MDPI adhere to. I think the author list should probably be shortened and those removed acknowledged instead.

Simple summary:

·       Internal vaginal ring is not correct terminology, either internal inguinal ring or more accurately just vaginal ring. It seems to be used interchangeably within the text and needs to be unified

·       A conservative management is not good grammar

·       I’m not sure background information defining barbed suture is necessary in a summary, “barbed suture” is descriptive enough and most people are familiar with this these days

Abstract:

·       Line 51- not necessary to repeat foals and adult horses, you’ve made this point clear

·       Line 54 – should say acquired I believe. Also bad grammar on this line – were did not

·       You’ve written using barbed sutures alone or combined with other techniques yet nothing results-wise about these other techniques

Introduction:

·       There’s no discussion of why the existing techniques are unsuitable and thus a new technique with barbed suture needs to be investigated.

Results:

·       Line 164 – 40 records of horses is ambiguous, I think you mean records of 40 horses

·       Line 208 – in the introduction you said the aim of the study was to assess the technique in a testis-sparing scenario, yet this horse got castrated…

·       Line 216 – probably prudent to mention what hernioplasty technique was used here given the study

·       Line 220-221 – you’ve gone from 28 to 30 cases…

·       Line 220-233 – this whole paragraph isn’t very clear and hard to keep track of as a reader

·       Table 1 – maybe a flowchart diagram would be more useful than this unusual table. Additionally, you’ve said in brackets the post-op herniations, yet you’ve used two types of brackets (square and round); this should be clarified

·       Line 242 – more castrations

·       Line 248 – 63 sparing testicles but 10 horses castrated in paragraph above, the maths doesn’t really add up or make sense here

·       Section 3.2 – the whole first paragraph is very confusing. I had to hand write out the words to try to see where all the cases came from and how they added up to 67, this needs to be clarified

·       Line 254 and Table 2 - seems to suggest 8 cases were included without barbed suture – again the aim of this study is to analyse barbed suture, yet there are 8 vaginal rings closed without barbed suture

·       Table 2 – not sure why it is included halfway through section 3.3.1 as it should be higher up.

·       Section 3.3 – is described as technique with barbed suture. There is no description of the non-barbed suture ones, which is fine as I think they should be excluded from the report. But from section 3.2, 13 had a combination of barbed suture and other techniques, 10 of which were cyanoacrylate. There is not description of these or what happened in the 3 other ones that weren’t cyanoacrylate. Was cyanoacrylate used to strengthen the barbed suture repair? Was one technique used on one side and a different one on the other?

·       Line 287-288 – in addition and additional – please improve English

·       Line 292 – it would be good to know what you are putting through your 4 portals. I understand 1 laparoscope, 1 suture device and you mention later in the manuscript about atraumatic forceps to manipulate the testicular cord, but not sure what a 4th might be for

·       Line 304-305 – what is HSRG (can’t find this on manufacturer website or anything) and “with a button in an extreme” makes no sense

·       Line 312 – referencing appears incorrect as 24 is do with a caecum

·       Figure 3 – a) not sure what is meant by the word aspect here

·       Section 3.3.3 – no mention of exercise or rehab plan or other instructions

·       Line 370 – “inguinal herniation post-hernioplasty were recorded” is not clear English

·       Section 3.4.1 and 3.4.3 – do you have any pictures of herniation post hernioplasty? No description if there was failure of the hernia repair or how the herniation occurred as in figures 2 & 3, your closure looks very tight, so hard to imagine reherniation

Discussion:

·       Line 408 – outcomes data is not good English

·       Line 413 - Additional costs related to implants – this probably needs expanding on as barbed suture is an implant with an increased cost and in 92.5% of your cases an automated suture device was used, which is not cheap or standard equipment either.

·       Line 418-419 – Two adults were operated under GA, however on line 274 it says one adult horse under GA?

·       Line 465 – commertial – incorrect spelling

·       Line 469 – hernioplastia – English

·       Line 473-474 – “could not be assessed” followed immediately by except…so you haven’t reported it when you apparently have the data – why would you not eliminate this limitation?

·       Line 483 – not sure what these spirals are, I assume they are supposed to be the degrees symbol ˚

·       Line 485 – last suturing device is not clear English. I assume you mean the SILS Stitch

·       Line 491 – associated with would be better English

·       Line 492 – should be presented with hernia recurrence

·       Line 491-506 – you’ve provided a comparison to the peritoneal flap technique and suggestions to avoid reherniation, but not comparison to other techniques that you have referenced in the paper

·       Figure 4 – not sure how much value this adds over the previous information and is rather similar to Figure 2 where only the craniolateral part was sutured. Also a) reads craniocaudal side, which is incorrect either way

·       Line 515 – Even although is not correct English

·       Line 519-520 – No new information should appear in the results/discussion that we weren’t warned about in the methods – this is the first place you’ve defined what a successful outcome was.

Conclusions:

·       Line 527-528 - Having decided not to look at the pre- and post-spermiograms, it’s hard to say minimal impact on reproductive fertility as you haven’t tested or proved that, other than some continuing their stallion careers

·       Line 532-533 – except testicles were not preserved in all cases

Comments on the Quality of English Language

See comments above - there are multiple instances where I have commented on the English of the paper

Author Response

General concepts:

I think this paper has information in it that deserves publication, however the overall impression of the paper is very convoluted and hard to track the details. I think there are ways that the presentation of the data could be improved and more clearly described.

We appreciate your valuable feedback and suggestions aimed at enhancing the manuscript. Your comments have been thoughtfully reviewed and incorporated into the document. Attached is the original manuscript showing the revisions made, highlighted using the track changes feature. Within this file, you will find our responses to each of your comments (where reviewer's comments are presented in bold and our responses in plain text).

I have a major problem with the case inclusion – the end of the introduction says “This study aims to assess the safety and effectiveness of a laparoscopic hernioplasty, with a focus on preserving the testicle, using knotless barbed sutures to close the VR in foals and adult horses and to provide a follow-up data on the outcome” and cases that were castrated and barbed suture were not used were included.

Thank you for highlighting this important aspect. As included in the discussion of the manuscript “This study aimed to describe and amplify the veterinary literature by presenting a wide range of clinical cases treated by using barbed sutures, including prophylactic, congenital and acquired inguinal hernias in several clinical conditions, such as incarcerated and strangulated hernias, in previously herniated inguinal rings or in contra-lateral inguinal rings (herniated or not).

Essentially, what we intended to convey in the introduction differs slightly from what was actually stated. Rather than accurately reflecting our objective, the initial wording implied that the cases discussed primarily involved instances where one or two testes needed preservation in animals. We have since revised this section for better clarity.

I think it would be OK to say 40 horses and 67 vaginal rings were included, of these 13 cases were castrated and 8 vaginal rings were closed without barbed suture – these cases were excluded, leaving X number of horses in the study.

After considering our response to the previous feedback, we believe that horses on which hernioplasty of a previously castrated ventral recumbency (VR) was performed, should be included as well. Among the 40 clinical cases included, barbed sutures, either alone or in conjunction with other techniques, were used in at least one VR. It is acknowledged that only 59 VRs, not 67 as previously stated, were treated with barbed sutures, with 46 being standalone and 13 in combination. We appreciate the identification of this discrepancy, which has now been rectified in the abstract, results, and previously Table 2 (now referred to as Table 1). Notably, excluding these 8 vaginal rings does not impact the total number of clinical cases, as barbed sutures were indeed employed in the other vaginal ring among these cases. Furthermore, the section discussing the surgical procedure in Results 3.2 has been revised and reorganized to enhance clarity.

Also 17 authors seems like a lot, if some are purely providing cases, that doesn’t automatically deserve authorship. From the author contribution statement, it does not seem that all authors qualify for the 4 criteria required by the ICMJE guidelines that MDPI adhere to. I think the author list should probably be shortened and those removed acknowledged instead.

We acknowledge that the quantity of co-authors may appear elevated compared to typical standards for this type of research. Nevertheless, it is crucial to underline that this study is conducted across multiple centres, necessitated by the need for a substantial volume of cases. Our selection criteria ensured that only individuals directly involved in the cases were appointed as co-authors, excluding mere observers. Furthermore, not all veterinarians participating in these surgical procedures have been listed as co-authors. Consequently, we express our gratitude to the personnel at the 12 hospitals where surgeries took place through acknowledgments.

All co-authors meet the 4 ICMJE criteria:

1.Substantial contributions to the conception or design of the work; or the acquisition, analysis, or interpretation of data for the work.

2.Drafting the work or reviewing it critically for important intellectual content.

3.Final approval of the version to be published.

4.Agreement to be accountable for all aspects of the work, ensuring that questions related to the accuracy or integrity of any part of the work are appropriately investigated and resolved.

Possibly, some confusion may have emerged as certain authors are listed only under the category of 'methodology' in the 'Author Contributions' section. This might be an oversight on our part as we had assumed their involvement encompassed both writing this segment of the manuscript (criteria 1 and 2). Furthermore, we interpreted the statement "All authors have read and agreed to the published version of the manuscript” in the final sentence of that paragraph to imply that all authors not only reviewed and approved the final version (criteria 3 and 4) but also critically assessed it and contributed substantial intellectual content (criterion 2). Nevertheless, if the reviewer or the editorial team deems it necessary for us to revise the 'Author Contributions' section, please inform us.

Simple summary:

Internal vaginal ring is not correct terminology, either internal inguinal ring or more accurately just vaginal ring. It seems to be used interchangeably within the text and needs to be unified

We acknowledge your feedback and have made the necessary corrections in the document. Moving forward, we consistently use the term 'vaginal ring (VR)' to refer to the internal inguinal ring. The term 'inguinal ring' is now reserved for instances pertaining to the external inguinal ring or both the internal and external inguinal rings.

A conservative management is not good grammar

Apologies for the mistake, the article 'a' has been eliminated to ensure correct grammar.

I’m not sure background information defining barbed suture is necessary in a summary, “barbed suture” is descriptive enough and most people are familiar with this these days

The journal's instructions for this section indicate: It is vitally important that scientists are able to describe their work simply and concisely to the public, especially in an open-access on-line journal.

We acknowledged that while practitioners, specialists, and professionals are already knowledgeable about this particular suture type, it could be beneficial to provide descriptions for the general public within the intended audience of this section.

Abstract:

Line 51- not necessary to repeat foals and adult horses, you’ve made this point clear

Eliminated.

Line 54 – should say acquired I believe. Also bad grammar on this line – were did not

Corrected or rewritten.

You’ve written using barbed sutures alone or combined with other techniques yet nothing results-wise about these other techniques

A new paragraph has been included in results on these cases (lines 364-379).

Introduction:

There’s no discussion of why the existing techniques are unsuitable and thus a new technique with barbed suture needs to be investigated.

This had been briefly discussed in the Discussion (lines 476 to 478). But in response to the reviewer's comment a new sentence has been added to the introduction

Results:

Line 164 – 40 records of horses is ambiguous, I think you mean records of 40 horses

Corrected

Line 208 – in the introduction you said the aim of the study was to assess the technique in a testis-sparing scenario, yet this horse got castrated…

This has already been clarified in a previous commentary and the inclusion of such cases has been justified.

Line 216 – probably prudent to mention what hernioplasty technique was used here given the study

Regrettably, an outside surgeon performed the procedure in this instance, and no information regarding the employed surgical technique is accessible.

Line 220-221 – you’ve gone from 28 to 30 cases…

The paragraph has been revised in its entirety in response to the reviewer's feedback, which states that among the 30 cases displaying signs of colic, 28 are attributable to acquired hernias while the remaining two are congenital cases.

Line 220-233 – this whole paragraph isn’t very clear and hard to keep track of as a reader

This paragraph has been rewritten.

Table 1 – maybe a flowchart diagram would be more useful than this unusual table. Additionally, you’ve said in brackets the post-op herniations, yet you’ve used two types of brackets (square and round); this should be clarified

Thank you for this suggestion. We have removed the Table 1 and replaced it with the suggested flowchart (now Figure 2).

Line 242 – more castrations

This has already been clarified in a previous commentary and the inclusion of such cases has been justified.

Cases where the affected testicle was castrated during the episode of colic due to inguinal hernia were also included, as hernioplasty of the contralateral testicle on that side is performed to avoid possible herniation on the opposite side. A sentence has been added in the text to explain this.

Line 248 – 63 sparing testicles but 10 horses castrated in paragraph above, the maths doesn’t really add up or make sense here

This entire section has been re-written.

Section 3.2 – the whole first paragraph is very confusing. I had to hand write out the words to try to see where all the cases came from and how they added up to 67, this needs to be clarified

This entire section has been re-written.

Line 254 and Table 2 - seems to suggest 8 cases were included without barbed suture – again the aim of this study is to analyse barbed suture, yet there are 8 vaginal rings closed without barbed suture

This entire section has been re-written and the Table 2 (now Table 1) modified.

Table 2 – not sure why it is included halfway through section 3.3.1 as it should be higher up.

It has been moved up in the text.

Section 3.3 – is described as technique with barbed suture. There is no description of the non-barbed suture ones, which is fine as I think they should be excluded from the report. But from section 3.2, 13 had a combination of barbed suture and other techniques, 10 of which were cyanoacrylate. There is not description of these or what happened in the 3 other ones that weren’t cyanoacrylate. Was cyanoacrylate used to strengthen the barbed suture repair? Was one technique used on one side and a different one on the other?

A new paragraph has been included in results on these cases (lines 364-379).

Line 287-288 – in addition and additional – please improve English

Reworded

Line 292 – it would be good to know what you are putting through your 4 portals. I understand 1 laparoscope, 1 suture device and you mention later in the manuscript about atraumatic forceps to manipulate the testicular cord, but not sure what a 4th might be for

A new phrase was included to explain that (lines 335-340)

Line 304-305 – what is HSRG (can’t find this on manufacturer website or anything) and “with a button in an extreme” makes no sense

The term ‘button’ has been replaced by ‘stopper’ to refer to the stopper at the end of these sutures and the commercial reference has been appropriately indicated.

Line 312 – referencing appears incorrect as 24 is do with a caecum

Corrected. Sorry by the mistake

Figure 3 – a) not sure what is meant by the word aspect here

Reworded (now in Figure 4)

Section 3.3.3 – no mention of exercise or rehab plan or other instructions

The following sentence has been included: “Following the surgery, generally, in most of the horses, two weeks of regulated physical activity was prescribed for each horse, involving exclusively 10-20 minutes of hand walking per day. Subsequently, a gradual return to regular exercise levels was initiated, during which the majority of the horses were able to resume exercise at the same intensity as before the surgical intervention after a few weeks”.

Line 370 – “inguinal herniation post-hernioplasty were recorded” is not clear English

Rewritten

Section 3.4.1 and 3.4.3 – do you have any pictures of herniation post hernioplasty? No description if there was failure of the hernia repair or how the herniation occurred as in figures 2 & 3, your closure looks very tight, so hard to imagine reherniation

Regrettably, the urgency of treating inguinal hernia colic led to these cases being promptly addressed. The emergencies were managed by veterinarians not associated with this study, and as a result, no laparoscopic images of the ventral recurrences (VRs) were captured to elucidate the re-herniation process. Obtaining such images would have been particularly valuable since, as previously mentioned, the VRs are typically securely sutured. However, in emergency scenarios where laparoscopy is deemed unnecessary for colic treatment, it becomes challenging to justify its use.

Discussion:

Line 408 – outcomes data is not good English

Corrected

Line 413 - Additional costs related to implants – this probably needs expanding on as barbed suture is an implant with an increased cost and in 92.5% of your cases an automated suture device was used, which is not cheap or standard equipment either.

A new phrase was included expanding that (lines 481-484).

Line 418-419 – Two adults were operated under GA, however on line 274 it says one adult horse under GA?

Apologies for the error. The only adult horse that underwent surgery under general anaesthesia was not the source of confusion. This particular case involved a 6-month-old foal treated for a non-congenital acquired inguinal hernia. We appreciate your attention to detail in noting this discrepancy, and the correction has been made in the text. Thank you.

Line 465 – commertial – incorrect spelling

Changed to commercial

Line 469 – hernioplastia – English

Changed to hernioplasty

Line 473-474 – “could not be assessed” followed immediately by except…so you haven’t reported it when you apparently have the data – why would you not eliminate this limitation?

These spermiograms have already been mentioned in section 3.4.4. The phrase has been rewritten.

Line 483 – not sure what these spirals are, I assume they are supposed to be the degrees symbol ˚

That is indeed the case. We don't know why the symbol has been changed. We have corrected it.

Line 485 – last suturing device is not clear English. I assume you mean the SILS Stitch

That is indeed the case. We have corrected it.

Line 491 – associated with would be better English

Corrected

Line 492 – should be presented with hernia recurrence

Rewritten

Line 491-506 – you’ve provided a comparison to the peritoneal flap technique and suggestions to avoid reherniation, but not comparison to other techniques that you have referenced in the paper

This new sentence was added: No cases of recurrence were documented with other published surgical techniques used for laparoscopic RV closure in horses, except for the peritoneal flap approach mentioned above.

Figure 4 – not sure how much value this adds over the previous information and is rather similar to Figure 2 where only the craniolateral part was sutured. Also a) reads craniocaudal side, which is incorrect either way

This figure has been deleted

Line 515 – Even although is not correct English

Corrected

Line 519-520 – No new information should appear in the results/discussion that we weren’t warned about in the methods – this is the first place you’ve defined what a successful outcome was.

Sorry, the inclusion of this sentence was an error. It has been removed.

Conclusions:

Line 527-528 - Having decided not to look at the pre- and post-spermiograms, it’s hard to say minimal impact on reproductive fertility as you haven’t tested or proved that, other than some continuing their stallion careers

The lack of inclusion of the term 'fertility' in our conclusion stems from its incomplete evaluation. Nonetheless, we believe that the spermiograms of the 3 animals referenced in section 3.4.4 and the continued offspring production by the stallions support the assertion that the technique minimally affects reproductive functionality. To qualify the conclusion further, a modification has been made to the statement. Additionally, a new sentence recommends that future studies geared towards assessing the reproductive and fecundity impacts on stallions treated with this technique across a larger sample size are imperative.

Line 532-533 – except testicles were not preserved in all cases

Certainly, here is a rephrased version of the sentence you provided: "Indeed, we concur, which is why we have included the term 'especially'. Given that this signifies a conclusion, an effort has been made to generalize. Our findings demonstrate the efficacy of this approach in instances where the testes are conserved, thus justifying our decision to incorporate this conclusion.

Comments on the Quality of English Language

See comments above - there are multiple instances where I have commented on the English of the paper

Reviewer 2 Report

Comments and Suggestions for Authors

I think this is an excellent paper which introduces some novel techniques, associated complications and ventual outcomes.  The figures are very good and easy to interpret.  The authors have been conservative in their recommendations and interpretations.  The manuscript is well written and very well constructed.

Author Response

I think this is an excellent paper which introduces some novel techniques, associated complications and eventual outcomes.  The figures are very good and easy to interpret.  The authors have been conservative in their recommendations and interpretations.  The manuscript is well written and very well constructed.

Thank you for your feedback and comments.

Round 2

Reviewer 1 Report

Comments and Suggestions for Authors

Thank you for your responses. This paper is significantly improved and reads much more clearly now. 

Thank you for the authorship clarification, it was the methodology part that I thought was problematic as that might have been just people performing the surgery and final approval alone is not a qualification as the ICJME guidelines are AND statements for the 4 criteria, not OR. Your answer to this suggests that all guidelines are satisfied. 

I just have a couple of minor comments on the new material included in the revised draft:

Line 270: typo "barded"

Line 347: I still think "in an extreme" is confusing English. Saying a button/stopper at one of the suture line, would probably be clearer

Line 366: typo "RV" instead of "VR"

Line 370: comma and full stop, hard to tell if these sentences are supposed to lead into each other

Line 370: conventional surgery - what do you mean by this - barbed suture?

Author Response

REVIEWER 1 COMMENT:

Thank you for your responses. This paper is significantly improved and reads much more clearly now.

Thank you for the authorship clarification, it was the methodology part that I thought was problematic as that might have been just people performing the surgery and final approval alone is not a qualification as the ICJME guidelines are AND statements for the 4 criteria, not OR. Your answer to this suggests that all guidelines are satisfied.

AUTHORS RESPONSE:

Thank you. We have specified it in the section on Author Contributions (lines 579-580).

REVIEWER 1 COMMENT:

I just have a couple of minor comments on the new material included in the revised draft:

Line 270: typo "barded"

AUTHORS RESPONSE:

Corrected (line 253)

REVIEWER 1 COMMENT:

Line 347: I still think "in an extreme" is confusing English. Saying a button/stopper at one of the suture line, would probably be clearer

AUTHORS RESPONSE:

Reworded (lines 319-320).

REVIEWER 1 COMMENT:

Line 366: typo "RV" instead of "VR"

AUTHORS RESPONSE:

Corrected (line 340)

REVIEWER 1 COMMENT:

Line 370: comma and full stop, hard to tell if these sentences are supposed to lead into each other

AUTHORS RESPONSE:

Corrected (line 344).

REVIEWER 1 COMMENT:

Line 370: conventional surgery - what do you mean by this - barbed suture?

AUTHORS RESPONSE:

Rewrited (line 344)